# Calcium salts of long-chain fatty acids from linseed oil decrease methane production by altering the rumen microbiome *in vitro*

Yoshiaki Sato[1], Kento Tominaga[2], Hirotatsu Aoki[1], Masayuki Murayama[3], Kazato Oishi[1], Hiroyuki Hirooka[1], Takashi Yoshida[2], Hajime Kumagai[1]*

1 Division of Applied Biosciences, Laboratory of Animal Husbandry Resources, Graduate School of Agriculture, Kyoto University, Kyoto, Japan, 2 Division of Applied Biosciences, Laboratory of Marine Microbiology, Graduate School of Agriculture, Kyoto University, Kyoto, Japan, 3 Project Planning & Development Department, Taiyo Yushi Corp., Yokohama, Kanagawa, Japan

* hkuma@kais.kyoto-u.ac.jp

**Data Availability Statement:** All sequence data are available from the DDBJ database (accession number DRA010200.)

## Abstract

Calcium salts of long-chain fatty acids (CSFA) from linseed oil have the potential to reduce methane ($CH_4$) production from ruminants; however, there is little information on the effect of supplementary CSFA on rumen microbiome as well as $CH_4$ production. The aim of the present study was to evaluate the effects of supplementary CSFA on ruminal fermentation, digestibility, $CH_4$ production, and rumen microbiome *in vitro*. We compared five treatments: three CSFA concentrations—0% (CON), 2.25% (FAL) and 4.50% (FAH) on a dry matter (DM) basis—15 mM of fumarate (FUM), and 20 mg/kg DM of monensin (MON). The results showed that the proportions of propionate in FAL, FAH, FUM, and MON were increased, compared with CON ($P < 0.05$). Although DM and neutral detergent fiber expressed exclusive of residual ash (NDFom) digestibility decreased in FAL and FAH compared to those in CON ($P < 0.05$), DM digestibility-adjusted $CH_4$ production in FAL and FAH was reduced by 38.2% and 63.0%, respectively, compared with that in CON ($P < 0.05$). The genera *Ruminobacter*, *Succinivibrio*, *Succiniclasticum*, *Streptococcus*, *Selenomonas.1*, and *Megasphaera*, which are related to propionate production, were increased ($P < 0.05$), while *Methanobrevibacter* and protozoa counts, which are associated with $CH_4$ production, were decreased in FAH, compared with CON ($P < 0.05$). The results suggested that the inclusion of CSFA significantly changed the rumen microbiome, leading to the acceleration of propionate production and the reduction of $CH_4$ production. In conclusion, although further in vivo study is needed to evaluate the reduction effect on rumen $CH_4$ production, CSFA may be a promising candidate for reduction of $CH_4$ emission from ruminants.

## Introduction

Methane ($CH_4$) is an important global greenhouse gas because it has a global warming potential 28 times as strong as that of carbon dioxide ($CO_2$) over a 100 years timeframe [1]. Livestock are the largest emitter of anthropogenic $CH_4$ and the global emission of $CH_4$ from

**Funding:** This study was supported by a Grant-in-Aid for the Japan Society for the Promotion of Science (JSPS) Fellows (20J15021) and a Grant from GAP Fund Program of Kyoto University (2019).

**Competing interests:** Masayuki Murayama was employed by Taiyo Yushi Co. that supplied CSFA for the study. All the other authors, including Mr. Masayuki Murayama, declare no competing interests. This does not alter our adherence to PLOS ONE policies on sharing data and materials.

livestock production was estimated as 195 Tg/year in 2003–2012 [2]; $CH_4$ released from enteric fermentation of ruminants accounts for 39% of $CH_4$ from livestock sector [3]. Methane is the end product of anaerobic fermentation in the digestive process of ruminants, contributing an energetic loss of 2–12% of the gross energy [4]. Therefore, mitigating enteric $CH_4$ emission from ruminants is required not only for reducing the environmental load but for improving the efficiency of animal production.

Dietary supplementation of lipids or independent fatty acids (FA) is one of the feasible feeding strategies to mitigate enteric $CH_4$ emission from ruminants [5–7]. Beauchemin et al. [5], through a meta-analysis, demonstrated that $CH_4$ production from ruminants was decreased by 5.6% with each 1% addition of supplemental fat. Among fats, polyunsaturated fatty acids (PUFA) are especially able to depress ruminal methanogenesis. Martin et al. [8] demonstrated that a 5.7% supply of linseed oil that includes a high proportion of PUFA significantly reduced $CH_4$ emitted from dairy cows by 64% *in vivo*. The reduction of enteric $CH_4$ production from ruminants in response to dietary fats or FA is due to their toxic effects against a wide variety of rumen microorganisms, including bacteria, protozoa, archaea, and fungi [9–13]. However, dietary lipids or FA also cause the reduction of other traits such as dry matter (DM) intake and nutrient digestibility [6, 8, 10], as well as $CH_4$ production.

Calcium salts of long-chain fatty acids (CSFA) have been widely used in dairy and beef production as a rumen-protected fat in practical farm conditions [14, 15]. Although dietary unprotected lipids significantly inhibit rumen microorganism activity, CSFA prevents problems related to rumen microbial fermentation and digestion. [14]. As a result, dietary CSFA have generally no or little adverse effect on nutrient digestibility in ruminants [15–20]. Furthermore, CSFA partially escapes biohydrogenation (BH) of fatty acids by rumen microbes. Wu et al. [21] reported that net BH of total unsaturated $C_{18}$ in diets with added CSFA and animal-vegetable blend fat were 57.3% and 87.2%, respectively, in dairy cows. Therefore, dietary CSFA can effectively increase unsaturated fatty acids contents in cow's lower digestive tract, increasing meat quality such as linoleic acid concentration [15], and milk yield and quality [19, 20, 22].

Recently, the effect of CSFA on $CH_4$ production emitted from ruminants has attracted considerable interest. For example, Kliem et al. [23] reported that diets with the addition of 2.2 g oil/kg DM as CSFA from palm and linseed oil decreased $CH_4$ production in dairy cows. This is probably because unsaturated fatty acids in CSFA were not completely protected from dissociation [24], and were slowly released as free fatty acids in the rumen, influencing rumen microorganisms involved in $CH_4$ production. Nevertheless, the effects of CSFA on rumen microbiome have been little reported, and the impact of graded level of dietary CSFA on rumen $CH_4$ production is unclear. Therefore, the objective of the present study was to evaluate the effects of supplementary CSFA on *in vitro* ruminal fermentation, digestibility, $CH_4$ production and ruminal microbiome by comparing with those of fumarate and monensin that are major inhibitors of enteric $CH_4$ emission from ruminants [25–28]. In the present study, we hypothesize that the FA may be gradually released from CSFA in the rumen and alter the microbiome, inhibiting $CH_4$ production with little negative effect on rumen fermentation.

## Materials and methods

The experiment was approved by the Kyoto University Animal Ethics Committee (Permit Number: 31–33) and performed at the Graduate school of Agriculture, Kyoto University from July to August 2019. The CSFA used in the present study was received from Taiyo Yushi Corp., a Japanese commercial chemical manufacturer. The product contained 56.7% linseed oil and 27.6% silica gel as the fatty acids absorbent. The molar ratio of FA to calcium in CSFA

was adjusted to 2.8. The FA were constituted with 5.5% palmitic acid (C16:0), 0.1% palmitoleic acid (C16:1), 3.3% stearic acid (C18:0), 18.2% oleic acid (C18:1), 15.6% linoleic acid (C18:2), 56.8% α-linolenic acid (C18:3) and 0.5% other fatty acids. Rolled barley was used as a substrate in the study. The substrate was ground in a Wiley mill to pass a 1 mm screen before use.

### Experimental design

The following five treatments (FAL, FAH, FUM, MOM, and CON) were used in the experiments. CSFA was supplemented at 2.25% DM and 4.50% DM of the substrate—namely FAL and FAH, respectively. Based on the linseed oil concentration of the CSFA used in this study, the linseed oil concentration in FAL and FAH were 1.5% and 3.0%, respectively. Fumarate was added to a final concentration of 15 mM (FUM). One treatment received monensin at 20 mg/kg DM of the substrate (MOM). The doses of fumarate and monensin were determined based on Shirohi et al. [29] and Joyner et al. [30]. The control treatment (CON) contained only substrate. Monensin was dissolved in ethanol before adding to test tubes in MON. Therefore, an equal volume of ethanol, 14.9 μL, was added into the other test tubes.

### Animals, diets, and feeding

Two ruminal-cannulated Corridale wethers with initial body weight (BW) of 58.6 ± 6.2 kg (mean ± SD) were used. The animals fed on ryegrass straw and concentrate at a ratio of 30:70 on a DM basis for 23 days. The amount of total diets provided was 2% of BW on a fresh matter (FM) basis in two equal portions daily, at 08:30 and 17:00. The ingredient compositions of the concentrate were as follows: 35.2% rice bran, 54.0% rolled barley, 6.9% alfalfa meal, 3.4% soybean meal, and 0.6% vitamin-mineral premix on a DM basis calculated using Standard Tables of Feed Composition in Japan [31]. Mineral blocks and water were offered *ad libitum*.

### Procedure of *in vitro* experiment

On day 24, about 200 mL ruminal fluid was collected through the rumen cannula from each wether before morning feeding and was transferred to the laboratory within 30 min. The sample was filtered through four layers of cheesecloth. Subsequently, the two strained liquids were mixed equally. The filtered sample were mixed with artificial saliva [32] in a ratio of 1:4 under anaerobic condition. The artificial saliva was sterilized by autoclaving and made anaerobic by a $CO_2$ flushing before mixing. A 40 mL mixture was transferred to each test tube containing 0.5 g DM of rolled barley and respective feed additives. The test tube was closed with a silicone rubber stopper fitted with a plastic syringe [33] to collect fermentation gas and incubated at 39˚C for 48 h. Each treatment was set up in three replicates.

During incubation, the total cumulative gas production at 0, 3, 6, 9, 12, 18, 24, 30, 36, 42, and 48 h, and $CH_4$ and $CO_2$ production at 12 and 24 h were measured. After incubation, test tubes were placed in ice-cold water to stop fermentation and immediately analyzed for pH using a pH meter (Horiba Ltd., Kyoto, Japan). Next, 1.5 mL of culture was subsampled for microbiome analysis and stored at -80˚C until further use. A 0.5 mL of the culture was mixed with 4.5 mL methyl green formalin sodium chlorate (MFS) solution for protozoa count [34]. All of the remaining culture was then centrifuged at $500 \times g$ for 5 min to separate the residue and supernatant. The supernatant was mixed with 25% (w/v) meta-phosphoric acid at a 5:1 ratio and stored at -20˚C until the analyses of volatile fatty acids (VFA) and ammonia nitrogen ($NH_3$-N) concentrations. The residue was transferred to a nylon bag to determine the digestibility of DM and neutral detergent fiber expressed exclusive of residual ash (NDFom).

## Chemical analyses

DM, crude protein (CP), ether extract (EE), and crude ash contents of the feeds and substrate were analyzed according to the standards of the Association of Official Analytical Chemists (AOAC 2000; 930.15, 976.05, 920.39, and 942.05, respectively). The NDFom and acid detergent fiber expressed exclusive of residual ash (ADFom) contents were determined according to Van Soest et al. [35]. The content of non-fibrous carbohydrate (NFC) was calculated using the following formula; NFC = 100 - (CP + EE + NDFom + crude ash). Chemical compositions of the feeds and substrate are shown in Table 1. The DM and NDFom digestibility were determined by the procedure described by Sato et al. [36]. The total $CH_4$ and $CO_2$ production were analyzed by gas chromatography (INORGA, LC Science, Nara, Japan) equipped with a thermal conductivity detector (TCD). For the analysis of VFA concentrations, collected samples were centrifuged at $15,000 \times g$ at 4°C for 15 min. The concentrations of VFA in the supernatants were determined by gas chromatography (GC14-B, Shimadzu, Kyoto, Japan) equipped with a FID using a packed glass column (Thermon 3000–2% Shimalite TPA 60/80 3.2 mmφ × 2.1, Shimadzu Co., Ltd., Kyoto, Japan). The temperature of injection, column, and detector were 250, 115, and 250°C, respectively. The $NH_3$-N concentration was determined by the steam distillation in a micro-Kjeldahl system (Kjeltec 2300, Foss Japan Ltd., Tokyo, Japan). Briefly, 3 mL of the supernatant after incubation was distilled with NaOH and the $NH_3$-N was trapped in a boric acid solution. Then, the solution was titrated with 0.1 N $H_2SO_4$ to determine $NH_3$-N concentration.

## Microbial DNA extraction, 16S rRNA gene amplicon preparation, and sequencing

Frozen culture samples were thawed on ice and centrifuged at $12,000 \times g$ for 15 min. The supernatant was removed, and the pellet was used for DNA extraction by the method reported by Frias-Lopez et al. [37]. Extracted DNA was stored at −20°C until further analysis. For each sample, the V3-V4 hypervariable region of the 16S rRNA gene was amplified using the following primer set reported by Takahashi et al. [38]; 341F (5′–CCTACGGGRSGCAGCAG–3′) and 805R (5′–GACTACCAGGGTATCTAAT–3′) added the Illumina overhang adapter sequences (forward: TCGTCGGCAGCGTCAGATGTGTATAAGAGACAG, reverse: GTCTCGTGGGCTCGGA GATGTGTATAAGAGACAG), according to the 16S sample preparation guide (https://support. illumina.com/content/dam/illumina-support/documents/documentation/chemistry_ documentation/16s/16s-metagenomic-library-prep-guide-15044223-b.pdf). The amplicons were then sequenced on Illumina MiSeq platform (Illumina, San Diego, CA, USA), which generated paired 300-bp reads.

**Table 1. Chemical compositions of feeds and substrate (% DM).**

| Item[1] | Concentrate | Ryegrass straw | Rolled barley |
|---|---|---|---|
| Dry matter (%) | 87.9 | 87.6 | 87.8 |
| Organic matter | 97.0 | 95.7 | 97.5 |
| Crude protein | 14.5 | 7.0 | 13.5 |
| Ether extract | 3.4 | 2.4 | 2.5 |
| NDFom | 33.4 | 64.5 | 32.1 |
| ADFom | 9.0 | 39.6 | 10.3 |
| Non-fibrous carbohydrate | 45.7 | 21.8 | 49.4 |
| Crude ash | 3.0 | 4.3 | 2.5 |

[1] NDFom, neutral detergent fiber expressed exclusive of residual ash; ADFom, acid detergent fiber expressed exclusive of residual ash.

## Sequence read processing and analysis

QIIME 2 (2019.4) package (http://qiime2.org) was used for sequence data analysis [39]. The adapter of the sequences was first trimmed using the cutadapt plugin [40]. The pair-end reads were then merged, quality filtered (Q20), and dereplicated using vsearch [41] and quality-filter plugin [42]. Subsequently, chimeras were identified and removed, and operational taxonomic units (OTUs) clustering using a similarity threshold of 97% were performed with the vsearch plugin [41]. Multiple sequence alignment of the sequences was performed using Multiple Alignment using Fast Fourier Transform (MAFFT) program [43] and masked [44] to remove highly variable regions using the qiime alignment command. A phylogenetic tree was then constructed with FastTree2 using the qiime phylogeny plugin [45]. The taxonomy of the sequence variants was assigned using the q2-feature-classifier plugin [46] against the Silva 132 OTUs sequences [47]. The OTUs were rarefied to a depth of 3,966, which was the lowest sample depth, for alpha and beta diversity analysis. For analysis of alpha diversity, richness (observed-OTUs and Chao1 [48]) and diversity (Shannon diversity index [49]) were estimated using the q2-diversity plugin. Non-metric multidimensional scaling (NMDS) ordination based on Bray–Curtis dissimilarities of OTUs was performed using R package 'vegan' [50] and visualized in R using 'ggplots2' [51]. Ward linkage hierarchical clustering using Spearman distance of OTUs was performed using the R function "hclust." In order to identify differentially abundant microbial taxa at the phylum and genus levels, we normalized the count matrices of taxa with a negative binomial distribution using DESeq2 [52]. Relative abundance was calculated using the normalized data, and the minor phylum and genus (average relative abundance < 1% for all treatments) were excluded from statistical analysis.

## Statistical analyses

Data, except for Bray–Curtis dissimilarities of OTUs and abundant bacterial taxa, were analyzed using GLM procedure of Statistical Analysis System (SAS, 2008). The mathematical model was:

$$Y_{ij} = \mu + T_i + e_{ij}$$

Where $\mu$ = the overall means, $T_i$ = the effect of treatment, and $e_{ij}$ = residual error. Multiple comparisons among the least square means were performed using the Tukey-Kramer method. In order to evaluate differences Bray–Curtis dissimilarities among the treatments, permutational multivariate analysis of variance (PERMANOVA) test was conducted with 9999 permutations using R package 'vegan' [50]. Differentially abundant bacterial taxa were identified using a negative binomial Wald test in DESeq2 [52]. The obtained p-values were corrected according to Benjamini and Hochberg procedure [53]. Differences were considered statistically significant at P < 0.05.

## Results

### *In vitro* gas production, CH₄ production, and nutrient digestibility

The effects of the feed additives on *in vitro* gas production, $CH_4$ production, and digestibility are shown in Table 2. Among the treatments, FUM had the highest total gas production (P < 0.05) at the time points of incubation investigated. Compared to CON, the total gas production at 12 h after incubation in FAL was higher (P < 0.05) and that in FAH was similar (P > 0.05) but the total gas productions at 48 h after incubation in FAL and FAH were lower (P < 0.05). The total gas production in MON was lower (P < 0.05) than that in CON in the time points of incubation investigated. The total $CH_4$ production after 12 h and 48 h

**Table 2. Effects of feed additives on *in vitro* gas production, CH$_4$ production and digestibility of concentrate as substrate.**

| Item[1] | Treatment[2] | | | | | SEM[3] |
|---|---|---|---|---|---|---|
| | CON | FAL | FAH | FUM | MON | |
| Gas production (mL/0.5gDM) | | | | | | |
| 12 h | 88.9$^c$ | 97.0$^b$ | 92.7$^{bc}$ | 103.8$^a$ | 82.0$^d$ | 1.23 |
| 24 h | 124.1$^b$ | 117.4$^c$ | 106.9$^d$ | 140.6$^a$ | 112.8$^d$ | 1.21 |
| 48 h | 134.9$^b$ | 124.1$^c$ | 111.3$^d$ | 154.5$^a$ | 120.9$^c$ | 1.47 |
| CH$_4$ production | | | | | | |
| Total CH$_4$ after 12 h incubation (mL/0.5gDM) | 6.3$^a$ | 4.7$^{bc}$ | 3.0$^d$ | 5.3$^{ab}$ | 4.1$^c$ | 0.23 |
| Total CH$_4$ after 48 h incubation (mL/0.5gDM) | 13.3$^a$ | 7.9$^b$ | 4.4$^c$ | 12.4$^a$ | 8.6$^b$ | 0.46 |
| Adjusted CH$_4$ after 48h incubation (mL/g IVDMD) | 32.4$^a$ | 20.0$^b$ | 12.0$^c$ | 30.3$^a$ | 21.1$^b$ | 1.15 |
| Adjusted CH$_4$ after 48h incubation (mL/g IVNDFD) | 123.7$^a$ | 80.6$^b$ | 46.6$^c$ | 115.5$^a$ | 80.7$^b$ | 5.10 |
| CO$_2$ production after 48h incubation (mL/0.5gDM) | 90.4$^b$ | 87.2$^{bc}$ | 80.7$^{bc}$ | 107.5$^a$ | 77.5$^c$ | 2.49 |
| Digestibility (%) | | | | | | |
| IVDMD | 82.3$^a$ | 79.1$^b$ | 74.2$^c$ | 81.8$^a$ | 81.9$^a$ | 0.42 |
| IVNDFD | 67.3$^a$ | 61.4$^b$ | 59.7$^b$ | 67.0$^a$ | 66.9$^a$ | 0.97 |

$^{abcd}$ LSMeans in a row with different superscripts significantly differ (P < 0.05).

[1] DM, dry matter; IVDMD, *in vitro* dry matter digestibility; IVNDFD, *in vitro* neutral detergent fiber digestibility.

[2] CON, non-supplementation; FAL, 2.25% DM calcium salt of long-chain fatty acid supplementation; FAH, 4.50% DM calcium salt of long-chain fatty acid supplementation; FUM, fumarate supplementation; MON, monensin supplementation.

[3] SEM, standard error of means.

incubation and digestibility-adjusted CH$_4$ in FAL, FAH, and MON were significantly lower than those in CON (P < 0.05), and the lowest CH$_4$ production was produced in FAH. No significant differences were observed for all parameters related to CH$_4$ production between CON and FUM (P > 0.05). The DM and NDFom digestibility in FAL and FAH were lower (P < 0.05) than those of the other treatments.

## Characteristics of rumen fermentation and protozoa population

The results of rumen fermentation and the protozoa population are presented in Table 3. The pH in FAL and FAH were similar to those in CON and MON (P > 0.05), and higher than that in FUM (P < 0.05). No differences were observed among the treatments for total VFA concentration, the proportion of iso-butyrate, n-butyrate, and iso-valerate. The percentages of acetate in FAL, FAH, and MON were lower (P < 0.05) than that in CON. In contrast, higher proportions of propionate were observed in FAL, FAH, FUM, and MON than in CON (P < 0.05). Lower ratios of acetate to propionate were observed in all additive treatments, compared to CON. NH$_3$-N concentration in FAL and FAH were lower than those in CON, FUM, and MON (P < 0.05). Compared with CON and FUM, smaller number of protozoa was observed in FAL (P < 0.05) and even fewer in FAH and MON (P < 0.05).

## Diversity and structure of rumen microbiome

The number of observed OTUs and Chao1 index in FAH and MON were lower (P < 0.05) than those in CON, while no differences were observed among CON, FAL, and FUM (Fig 1). For the Shannon diversity index, CON has the highest, followed by FAL and FUM, and FAH and MON showed the lowest values (P < 0.05) (Fig 1). PERMANOVA analysis confirmed that there were significant differences of rumen microbial communities among the treatments (P < 0.001), and NMDS using the Bray-Curtis dissimilarity metric (Fig 2) and hierarchical

**Table 3. Effects of feed additives on pH, NH$_3$-N, protozoa population and VFA after 48 h incubation.**

| Item[1] | Treatment[2] | | | | | SEM[3] |
|---|---|---|---|---|---|---|
| | CON | FAL | FAH | FUM | MON | |
| pH | 6.48$^a$ | 6.49$^a$ | 6.46$^a$ | 6.32$^b$ | 6.40$^{ab}$ | 0.02 |
| NH$_3$-N (mgN/dL) | 26.7$^a$ | 20.8$^c$ | 19.4$^c$ | 26.5$^a$ | 24.6$^b$ | 0.35 |
| Protozoa (×10$^5$/mL) | 4.3$^a$ | 3.1$^b$ | 2.0$^c$ | 4.4$^a$ | 1.9$^c$ | 0.19 |
| VFA | | | | | | |
| Total VFA (mmol/L) | 130.6 | 127.1 | 124.8 | 132.1 | 121.8 | 4.88 |
| Acetate (%) | 49.6$^a$ | 44.0$^b$ | 42.4$^b$ | 45.6$^{ab}$ | 44.9$^b$ | 0.91 |
| Propionate (%) | 35.2$^c$ | 40.7$^{ab}$ | 43.4$^a$ | 39.1$^b$ | 41.8$^{ab}$ | 0.63 |
| iso-Butyrate (%) | 0.22 | 0.00 | 0.00 | 0.11 | 0.00 | 0.06 |
| n-Butyrate (%) | 10.1 | 9.4 | 7.6 | 10.0 | 8.1 | 1.09 |
| iso-Valerate (%) | 2.6 | 2.9 | 2.1 | 2.9 | 2.2 | 0.24 |
| n-Valerate (%) | 2.3$^c$ | 3.1$^b$ | 4.5$^a$ | 2.3$^c$ | 3.0$^b$ | 0.14 |
| Acetate:Propionate | 1.4$^a$ | 1.1$^{bc}$ | 1.0$^c$ | 1.2$^b$ | 1.1$^{bc}$ | 0.03 |

$^{abcd}$ LSMeans in a row with different superscripts significantly differ (P < 0.05).

[1] NH$_3$-N, ammonia nitrogen; VFA, volatile fatty acids.

[2] CON, non-supplementation; FAL, 2.25% DM calcium salt of long-chain fatty acid supplementation; FAH, 4.50% DM calcium salt of long-chain fatty acid supplementation; FUM, fumarate supplementation; MON, monensin supplementation.

[3] SEM, standard error of means.

clustering of the microbiota community (Fig 3) revealed distinct clustering patterns that separated the microbiota in FAL, FAH and MON from that in CON and FUM.

## Bacterial abundance

Fig 4 shows the relative abundance of microbiota at the phylum level, and different abundant taxa is presented in S1 Table and Fig 5. At the phylum level, the microbiota in all treatments was dominated by Firmicutes, Bacteroidetes, and Proteobacteria. The abundant of the phyla Bacteroidetes in MON was lower than that in CON (P < 0.05). The abundant of the phyla Firmicutes in FAH, FUM and MON was increased compared with CON and FAL (P < 0.05), and that of Proteobacteria in FAL, FAH and MON was higher than that in CON (P < 0.05).

At the genus level, *Methanobrevibacter*, which accounted for over 99% of the phylum Euryarchaeota, in FAH was lower than that in CON (P < 0.05). Among the phylum Bacteroidetes, *Bacteroidales* BS11 gut group and *Rikenellaceae* RC9 gut group was higher in CON than FAL, FAH and MON (P < 0.05). There was significant difference of unclassified Bacteroidales between CON and the other treatments (P < 0.05). Regarding the phylum Firmicutes, many genera (*Succiniclasticum Anaerovibrio*, *Megasphaera*, *Schwartzia*, *Selenomonas.1*, *Veillonellaceae* UCG.001, uncultured *Veillonellaceae*, and unclassified *Veillonellaceae*) were higher in MON than CON (P < 0.05). Similarly, adding CSFA at high level (treatment FAH) increased *Succiniclasticum*, *Selenomonas.1* and *Megasphaera* compared to those of CON (P < 0.05). Additionally, *Streptococcus* was higher in FAH than in the other treatments (P < 0.05) and *Schwartzia* was increased in FUM (P < 0.05). *Ruminococcus.2* in all additive treatments, especially FAH and MON, were significantly decreased compared with that in CON (P < 0.05). Among the phylum Proteobacteria, *Ruminobacter* was lower in FUM than other treatments (P < 0.05), but higher in FAH and MON than in CON (P < 0.05). *Succinivibrio* was increased in FAL and FAH (P < 0.05) compared with that in CON and MON. *Pyramidobacter* (the phylum Synergistetes) was higher in MON than that in other treatments (P < 0.05).

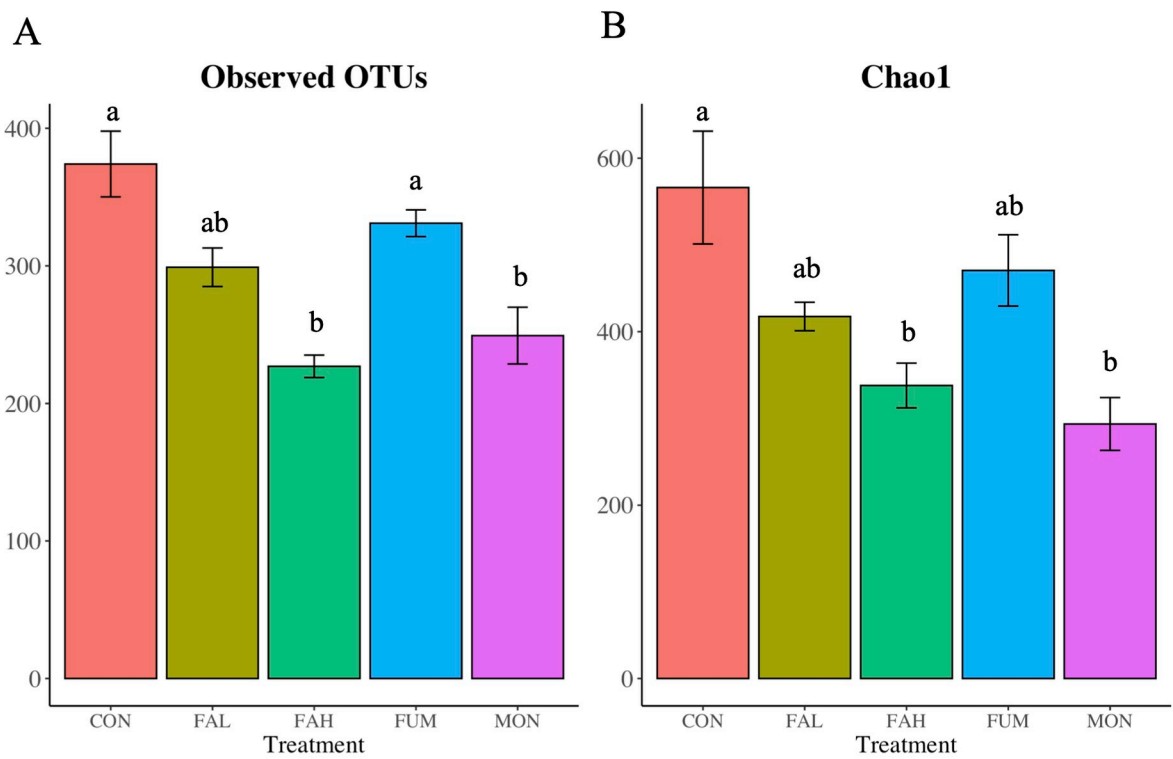

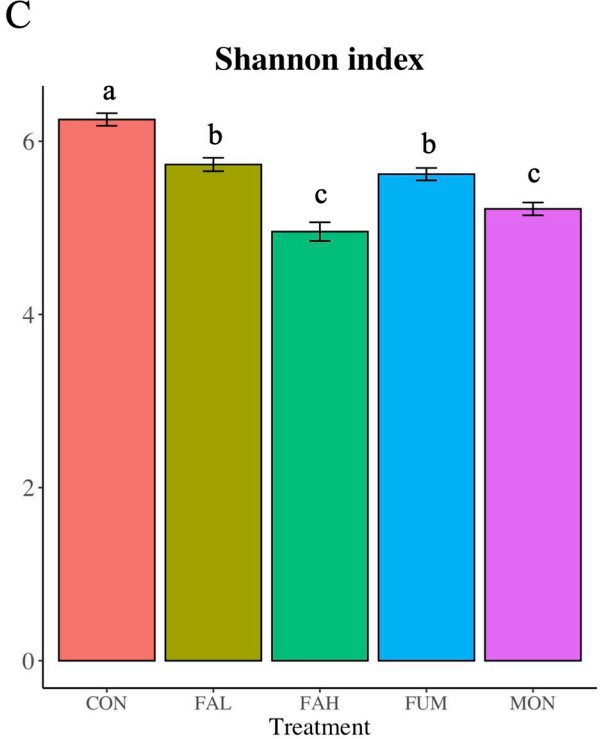

**Fig 1. Effects of feed additives on alpha diversity.** Data are presented as mean ± SE (n = 3 per treatment). (A) Observed OTUs, (B) Chao1, and (C) Shannon index in microbiomes after incubation. Different superscripts ([abc]) indicate significant differences (P < 0.05). CON = non-supplementation; FAL = 2.25% DM calcium salt of long-chain fatty acid supplementation; FAH = 4.50% DM calcium salt of long-chain fatty acid supplementation; FUM = fumarate supplementation; MON = monensin supplementation.

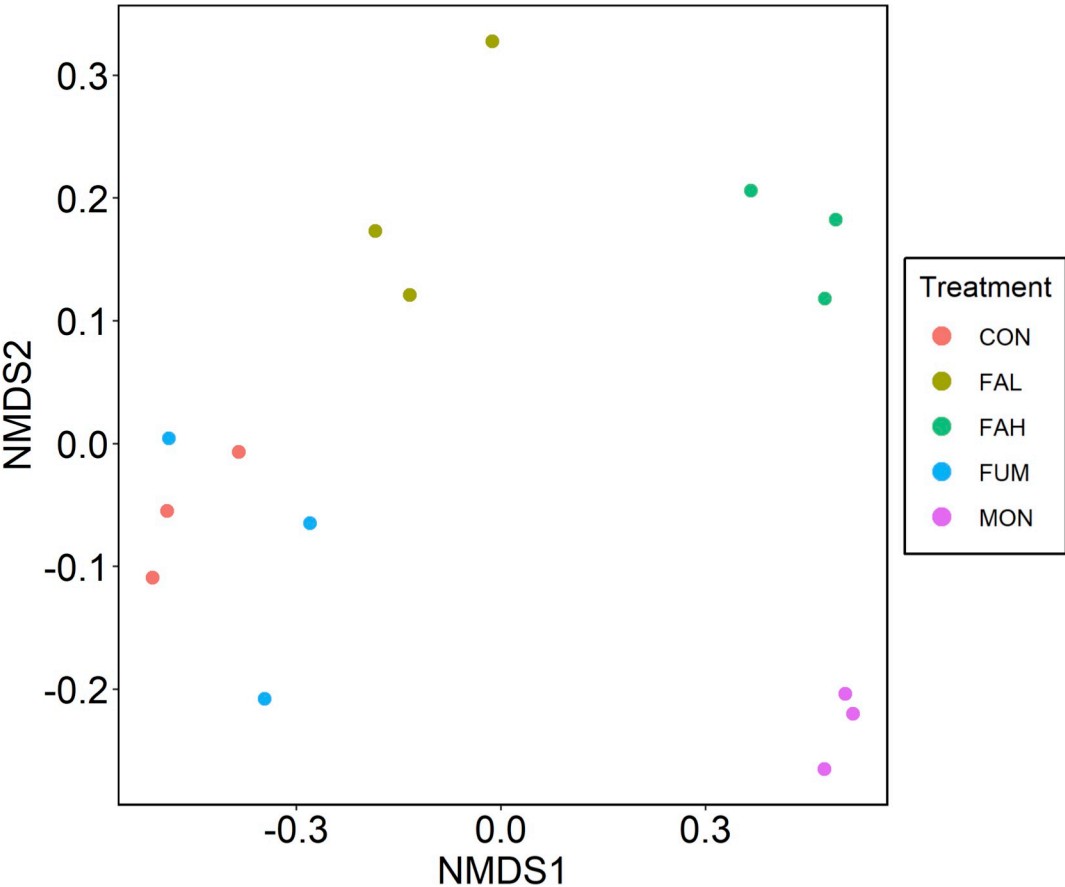

**Fig 2. Non-metric multidimensional scaling (NMDS) plots of the Bray-Curtis dissimilarities of microbiota.** CON = non-supplementation; FAL = 2.25% DM calcium salt of long-chain fatty acid supplementation; FAH = 4.50% DM calcium salt of long-chain fatty acid supplementation; FUM = fumarate supplementation; MON = monensin supplementation.

## Discussion

We evaluated the effect of CSFA on *in vitro* rumen fermentation, $CH_4$ production, digestibility, and rumen microbiota. Many studies showed that supplementary linseed decreases ruminal $CH_4$. A meta-analysis by Martin et al. [7] demonstrated that for each 1% addition of supplemental linseed, $CH_4$ production decreased by 4.8%. In the present study, compared with control (no additive), low and high amounts of CSFA supplementation (FAL and FAH) reduced $CH_4$ production (mL/g IVDMD) by 38.2% and 63.0%, respectively. We found that addition of CSFA led to 21.0–25.5% decreases per 1% of linseed oil addition. Thus, in this study, the percentage of $CH_4$ reduction due to CSFA supplementation was higher than that reported by Szumacher-Strabel et al. (10.1% reduction per 1% of linseed oil addition in *in vitro*) [54], indicating that the CSFA used in the present study has a substantially high reduction effect on $CH_4$ production. We presumed that silica might be a key factor to increasing the $CH_4$ reduction effect of CSFA. Shinkai et al. [55] reported that cashew nut shell liquid pellet with 40% silica powder has a larger reduction effect on $CH_4$ production than that with 11.3% silica powder and several ingredients. They hypothesized that the cashew nut shell liquid pellet with 40% silica powder easily diffuses in the rumen, leading to a remarkable decrease in $CH_4$ production [55]. Similarly, unsaturated fatty acids might diffuse from CSFA with 27.6% silica gel, and efficiently suppressed microbial activity related to $CH_4$ production.

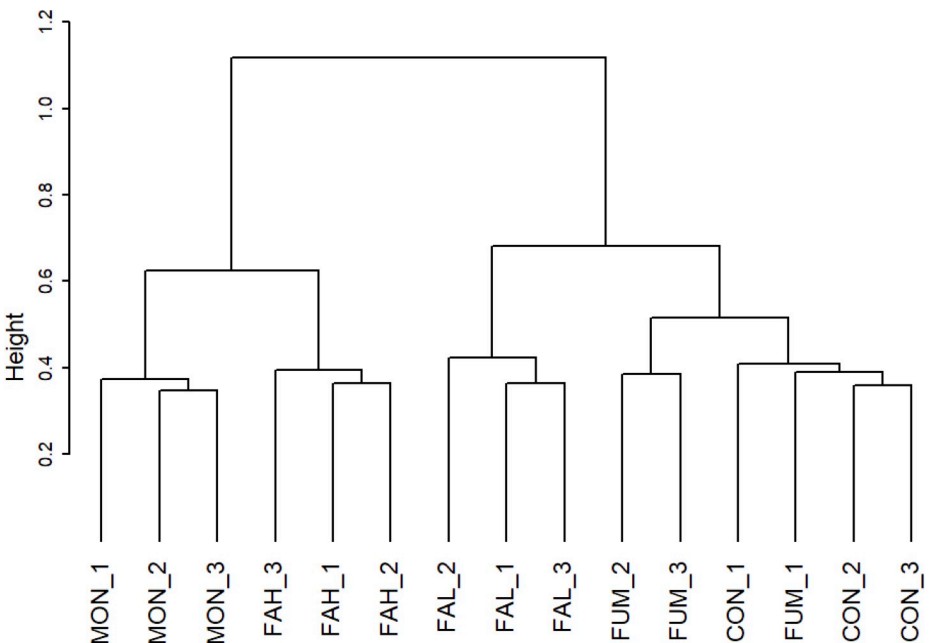

**Fig 3. Ward linkage hierarchical clustering of microbiota based on Spearman distance.** CON = non-supplementation; FAL = 2.25% DM calcium salt of long-chain fatty acid supplementation; FAH = 4.50% DM calcium salt of long-chain fatty acid supplementation; FUM = fumarate supplementation; MON = monensin supplementation; 1–3, sample number.

Furthermore, adding CSFA at a high dose level more clearly reduced $CH_4$ production when compared with adding supplementary fumarate and monensin. Monensin and fumarate are feed additives that can reduce $CH_4$ production from ruminants. Odongo et al. [26] reported that monensin reduced $CH_4$ production from dairy cows without the negative effect on DM intake and milk yield. Asanuma et al. [25] demonstrated that the use of fumarate as a feed additive could reduce methanogenesis and increase propionate production in the rumen, leading to the reduction of $CH_4$ production. Therefore, the results in the present study indicate that CSFA is one of the potent inhibitors of methanogenesis. In the present study, supplementary fumarate did not reduce $CH_4$ production probably because rolled barley was used as the substrate. García-Martínez et al. [56] reported that adding fumarate to batch culture under a low-forage substrate condition have less $CH_4$ reduction effect compared with a high-forage substrate condition.

Methane production in the rumen is due to methanogenesis of methanogens, and rumen methanogens use mainly $H_2$ to reduce $CO_2$ to $CH_4$ [57]. Protozoa, which produce $H_2$ in the hydrogenosomes [58], are also involved in methanogenesis because some of the methanogens attach to the cell surface of protozoa [59]. Guyader et al. [60] demonstrated by a meta-analysis that there was a positive linear correlation between protozoal numbers and $CH_4$ emissions. Fatty acids, especially PUFA, have adverse effects on methanogens and protozoa [61, 62]. In the present study, the genus *Methanobrevibacter*, which is the dominant methanogen in the rumen [63–65], and the count of protozoa were decreased with the levels of CSFA, suggesting that FA released from CSFA might influence these microorganisms.

Increasing propionate production decreases available $H_2$ for methanogenesis since propionate formation is a competing alternative to $H_2$ formation [66]. Therefore, the increase of propionate in the rumen is associated with reduction in $CH_4$ production. In the present study, the percentage of propionate was increased by CSFA supplementation, corresponding with the

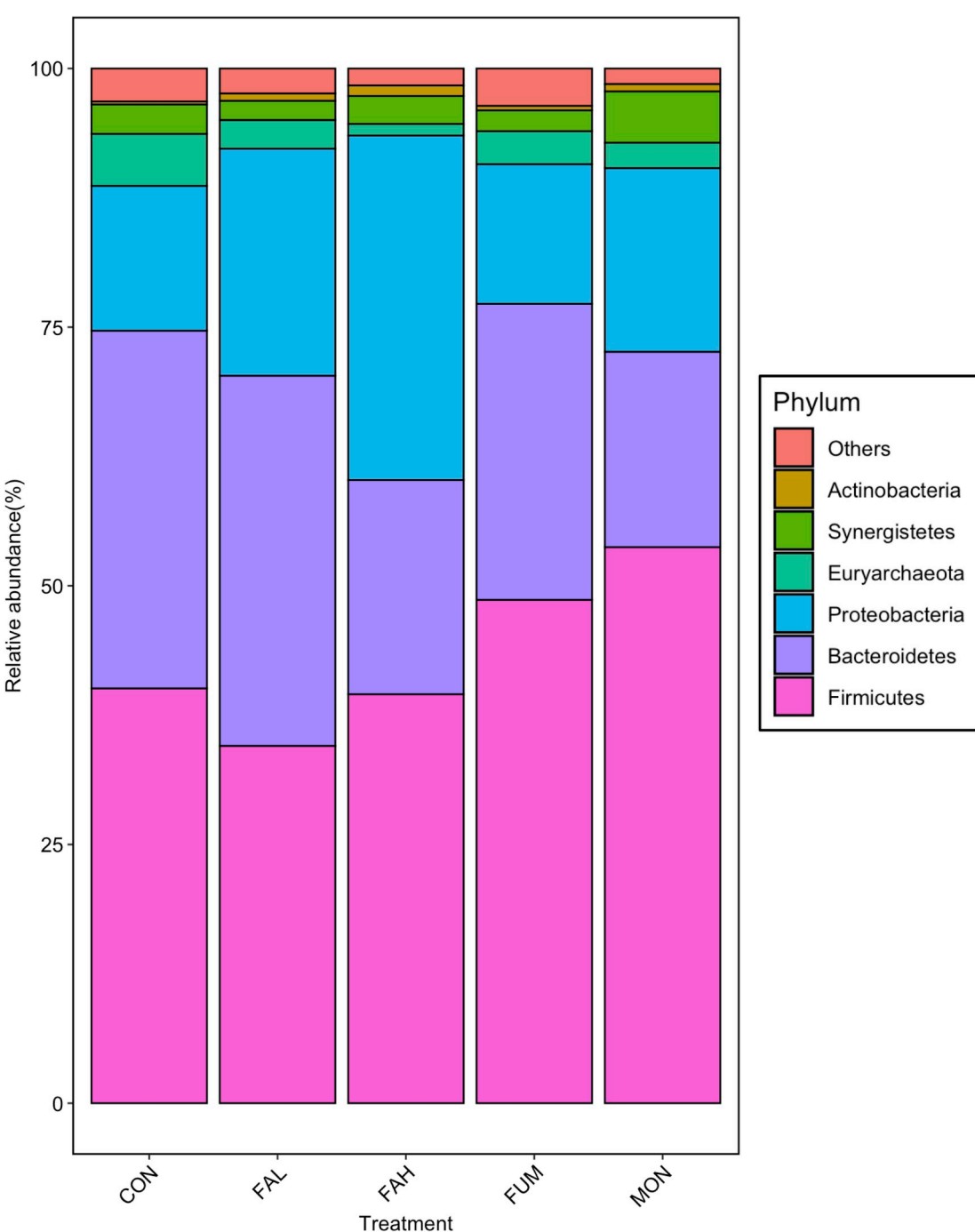

**Fig 4. Relative abundance (%) of rumen microbiome at phylum level.** All phyla comprising less than 1% of the total abundance in all treatments were combined into the "Others" category. CON = non-supplementation; FAL = 2.25% DM calcium salt of long-chain fatty acid supplementation; FAH = 4.50% DM calcium salt of long-chain fatty acid supplementation; FUM = fumarate supplementation; MON = monensin supplementation.

result of fumarate and monensin inclusion. These results are consistent with previous studies related to the supplementation of linseed oil in dairy cows and steers diets [67–69]. In the rumen, there are two pathways for propionate production; succinate pathway (the main pathway) and acrylate pathway [70]. In the succinate pathway, fumarate is reduced to succinate,

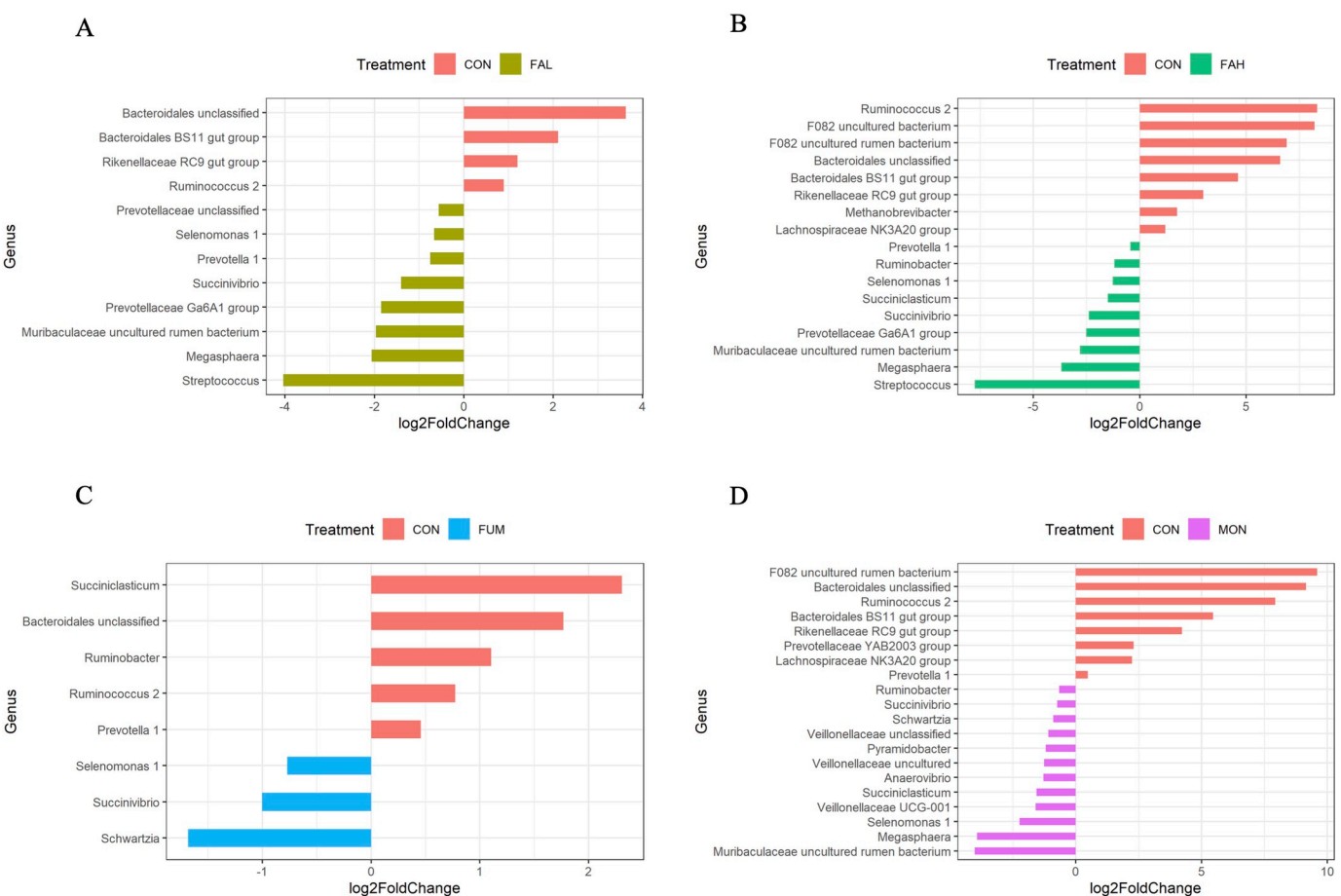

**Fig 5. Significantly differentially abundant microbial taxa at the genus level.** Genera with significant differences (adjusted P < 0.05) identified using DESeq2 between (A) CON and FAL, (B) CON and FAH, (C) CON and FUM, and (D) CON and MON. Only taxa ≥ 1% relative abundance for at least one treatment are shown. CON = non-supplementation; FAL = 2.25% DM calcium salt of long-chain fatty acid supplementation; FAH = 4.50% DM calcium salt of long-chain fatty acid supplementation; FUM = fumarate supplementation; MON = monensin supplementation.

and succinate is converted to propionate by some bacteria. The genera *Ruminobacter* [71] and *Succinivibrio* [72] are involved in succinate production, while *Succiniclasticum* [73], *Selenomonas* [74], and *Schwartzia* [74] ferment succinate and produce propionate via the succinate pathway in rumen. In the present study, the genera *Ruminobacter Succinivibrio*, and *Selenomonas.1*, *Succiniclasticum* were increased by supplementation of CSFA at a high level or monensin compared to control, indicating that *Ruminobacter* and *Succinivibrio* might produce succinate used by *Selenomonas* and *Succiniclasticum* for propionate production in CSFA or monensin supplementation. However, the inclusion of fumarate increased the genus *Schwartzia*. Thus, the main bacteria related to propionate production via succinate pathway were different between the treatments (FAH and MON) and fumarate although all feed additives increased the proportion of propionate. Furthermore, the genera *Streptococcus* and *Megasphaera* were also increased by the inclusion of CSFA at a high level. *Streptococcus bovis* produces lactate [75, 76], while *Megasphaera elsdenii* is a utilizer of lactate for the production of butyrate and propionate [75–77]. Thus, our results indicate that supplementary CSFA may also promote propionate production via acrylate pathway as well as succinate pathway.

Rumen protected fats such as CSFA prevent ruminal fermentation and digestion problems caused by fat feeding [14]. Therefore, we expected that no or little negative effects of

supplementary CSFA on rumen fermentation and digestibility would be observed as with the results of other studies [15–20]. In the present study, however, supplementary CSFA decreased DM and NDFom digestibility, resulting in the inhibition of total gas production after 48 h incubation. Decreased ruminal ammonia was also observed with CSFA inclusion. The results indicate that FA released from CSFA might be sufficiently detrimental to the activity of ruminal microorganisms. Yang et al. [11] reported that dietary soybean oil and linseed oil to dairy cows decreased the counts of cellulolytic bacteria. We observed a strong decrease in the genus *Ruminococcus*.2 (the family *Ruminococcaceae*), *Rikenellaceae RC9* gut group (the family *Rikenellaceae*), unclassified Bacteroidales (the order Bacteroidales), and *Bacteroidales* BS11 gut group (the family *Bacteroidaceae*) with the addition of CSFA. *Ruminococcus* is one of the main cellulolytic bacteria in the rumen, accounting for about $10^6$ cells/mL of rumen content [78]. Dai et al. [79] demonstrated that *Ruminococcus* primarily synthesized putative cellulases and hemicellulases. It is well known that long chain fatty acids inhibit the growth of gram-positive bacteria [80], and supplementary linseed oil reduces *Ruminococcaceae* [11, 81], which agrees with the results in the present study. *Rikenellaceae* may be associated with either primary or secondary degradation of structural carbohydrates [82]. Various studies have reported that supplementary oil such as sunflower oil [83], linseed oil [66], and tucumã oil [84] reduced the relative abundance of *Rikenellaceae* RC9, consistent with our findings. Some Bacteroidales are associated with fiber degradation. Bacteroidales BS11 is specialized in fermenting many different hemicellulosic monomers, producing acetate and butyrate for the host [85]. Hence, the reduced abundance of these bacterial taxa might be a reason for the decreased digestibility observed after CSFA supplementation. The inclusion of monensin also decreased these taxa because monensin preferentially inhibits gram-positive bacteria [86] as linseed oil. In contrast to CSFA, no reduction in fiber digestibility by monensin was observed. This may be probably due to higher abundance of some taxa which belong to the phylum Firmicutes in MON than in FAH. Bensoussan et al. [87] found that cellulosome components, which are an extracellular multi-enzyme complex considered to be one of the most efficient plant cell wall-degrading strategies, were prevalent in Firmicutes. Among the phylum Firmicutes, *Selenomonas*.1, which was significantly increased with monensin compared to FAH, might have enhanced fiber digestion in the present study. *Selenomonas ruminantium* improves fiber digestion by cooperating with other cellulolytic bacteria [88, 89].

Interestingly, cumulative gas production after supplementing CSFA at a low level was higher than in control at 12 h after incubation in spite of decreased $CH_4$ production. These results indicate that CSFA supplementation inhibits the activity of rumen microbes related to $CH_4$ production in the initial stage of ruminal fermentation without toxic effect on other rumen bacteria. The characteristic of CSFA may be worthy of *in vivo* investigation, since rumen contents and liquid flow out of the rumen *in vivo*. Hartnell and Satter [90] reported that ruminal turnover rates of liquid, grain, and hay were 8.1, 4.4, and 3.9% per hour, respectively, in dairy cows. Considering these turnover rates and our results, dietary CSFA may be able to decrease $CH_4$ production with little or no negative effect on rumen fermentation and digestibility *in vivo*.

One of the limitations of the present study was that the low sample size (n = 3 per treatment) with only one *in vitro* trial. Moreover, we evaluated the effect of CSFA using only one substrate although the effect of fat on rumen fermentation can be influenced by the concentrate and roughage ratio of feeds [91]. Therefore, further studies with an increase in sample size and substrates will be needed to increase the reliability of the effect of CSFA on $CH_4$ production.

In conclusion, although *in vitro* digestibility was reduced with increasing concentration of CSFA, addition of CSFA significantly changed rumen microbiome, resulting in the

acceleration of propionate production, and the reduction of $CH_4$ production. These findings present CSFA as a promising candidate for reduction of $CH_4$ emission from ruminants. However, some differences of the observation were reported between *in vivo* and *vitro* [92]. Therefore, future studies are needed to confirm the *in vivo* effect of dietary CSFA on $CH_4$ production, productivity, rumen microbiome, and digestibility, and to determine the optimal amount of CSFA in a diet for ruminants.

## Supporting information

**S1 Table. Differential abundance in specific taxa at phylum and genus level.** [1] CON, non-supplementation; FAL, 2.25%DM calcium salt of long-chain fatty acid supplementation; FAH, 4.50%DM calcium salt of long-chain fatty acid supplementation; FUM, fumarate supplementation; MON, monensin supplementation. [2] Phylum and genus exhibited significant differences (adjusted P < 0.05) identified using DESeq2 with $\geq$ 1% relative abundance in more than one treatment. [3] The p-value was adjusted using the Benjamini-Hochberg procedure. (DOCX)

## Author Contributions

**Conceptualization:** Yoshiaki Sato, Hajime Kumagai.

**Data curation:** Yoshiaki Sato.

**Formal analysis:** Yoshiaki Sato.

**Funding acquisition:** Yoshiaki Sato, Hajime Kumagai.

**Investigation:** Yoshiaki Sato, Kento Tominaga, Hirotatsu Aoki.

**Methodology:** Yoshiaki Sato, Hajime Kumagai.

**Resources:** Masayuki Murayama, Takashi Yoshida, Hajime Kumagai.

**Software:** Yoshiaki Sato, Kento Tominaga.

**Supervision:** Hajime Kumagai.

**Visualization:** Yoshiaki Sato.

**Writing – original draft:** Yoshiaki Sato.

**Writing – review & editing:** Yoshiaki Sato, Kazato Oishi, Hiroyuki Hirooka, Takashi Yoshida, Hajime Kumagai.

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
