## [Decision Letter · Decision Letter 0]

27 Aug 2020

PONE-D-20-14439

Calcium salts of long-chain fatty acids from linseed oil decrease methane production by altering rumen microbiome in vitro

PLOS ONE

Dear Dr. Kumagai,

Thank you for submitting your manuscript to PLOS ONE. After careful consideration, we feel that it has merit but does not fully meet PLOS ONE’s publication criteria as it currently stands. Therefore, we invite you to submit a revised version of the manuscript that addresses the points raised during the review process.

Both reviewers noted a number of issues that must be adequately addressed before further consideration can be made.

We look forward to receiving your revised manuscript.

Kind regards,

Brenda A Wilson, Ph.D.

Academic Editor

PLOS ONE

Journal Requirements:

'This study was supported by a Grant-in-Aid for the Japan Society for the Promotion of Science (JSPS) Fellows (20J15021) and a Grant from GAP Fund Program of Kyoto University (2019). '

We note that one or more of the authors are employed by a commercial company: Taiyo Yushi Corp

Reviewers' comments:

Reviewer's Responses to Questions

**Comments to the Author**

1. Is the manuscript technically sound, and do the data support the conclusions?

Reviewer #1: Partly

Reviewer #2: Yes

2. Has the statistical analysis been performed appropriately and rigorously? 

Reviewer #1: I Don't Know

Reviewer #2: Yes

3. Have the authors made all data underlying the findings in their manuscript fully available?

Reviewer #1: Yes

Reviewer #2: Yes

4. Is the manuscript presented in an intelligible fashion and written in standard English?

Reviewer #1: Yes

Reviewer #2: Yes

5. Review Comments to the Author

Reviewer #1: The current study evaluated the impact of Ca-salts of FA from linseed oil on in vitro fermentation characteristics. Although the practical application of feeding Ca-salts as a strategy to limit CH4 emissions is questionable mostly because it would just be cheaper to feed linseed oil vs Ca-salts (typically, the cost of Ca-salts > oil > oilseeds), and that linseed oil is possibly more effective than CSFA. However, if a reduction in emissions occurs with CSFA in vivo, it would be a great thing.

Major comments:

1. Although there was a dramatic decrease in CH4 production in the present study, it is questionable if similar observations would be made in vivo. This is related to the experimental approach. Besides being a function of the technology used (pH dependence for Ca-salts), the effectiveness of rumen-protection for products like CSFA is also influenced by factors including rumen residence time. Therefore, the in vitro system (batch culture for 48 h) is not ideal for testing the hypothesis. One could speculate that some of the observations made in the current study (e.g., greater reduction in CH4 production with FAL and FUM compared to previous in vivo studies with linseed oil, and decrease in DM and fiber digestion with FAL and FUM, which is not typically observed with CSFA in vivo) are all a result of the closed-system (no continuous addition of substrates or passage of digesta out; fixed residence time with incubation up to 48 h). It is highly likely that observations made in this study might not fully translate to in vivo conditions (continuous culture). Therefore, the comparison of observations made in this study to predominantly in vivo studies in the discussion, without accounting for the differences in approach (batch vs continuous culture), is problematic. The observations made should be interpreted with caution, and this should also be reflected in the wording of the concluding statement(s).

2. Given the day-to-day variation in both animal factors (e.g., nutrient intake and timing of meals, which can influence properties of the inoculum/rumen fluid) and human factors (e.g., possible errors when making reagents or purging of system to ensure anaerobic conditions, pipetting/measurement errors, etc), it is recommended that runs (days of incubation) are replicated. At least 2 incubation runs over 2 non-consecutive days are needed; in that case, the day effect is also tested in the statistical model, & is only removed if it is not significant. This potentially increases replicates (if there is no day effect), and thus experimental power, which was low in the present study, since only 3 replicates were used.

The importance of replicating runs is highlighted by the observed increase in GP at 12 h of incubation when CH4 decreased. One would have greater confidence that this observation was of biological significance as discussed, rather than an error in measurement/anomaly, if the same observation is made across runs/different days of incubation.

3. There is no indication that a negative control (artificial saliva + rumen fluid + substrate) was included in the in vitro run. Even if it was just 1 run, a “blank correction” is important for some measures including GP as it provides more meaning data compared to just absolute values.

4. With statistical analysis it is not clear if data was tested for normality and homoscedacity prior to analysis. This is especially important with bacterial relative abundance data (proportions).

Minor comments:

L4: change to “altering the rumen”

L37: change to “changed the rumen”

L39 – L41: This is highly speculative given the approach used (batch culture) in the present study.

L44: change to “because it has”

L46: change to “Livestock are the”

L49: delete “as a result”

L62: change to “due to their toxic’

L70: change to “CSFA have generally”

L91: change to “alter the microbiome”

L91: delete “resulting in”

L107 – L116: What was the actual experimental design? Was it a CRD? Information needs to be added

L109: replace “named” with “namely”

L110: change to “of the CSFA used in this study, the…”

L123 – L125: The ingredient composition should be reported on a DM basis.

L129: change to “through the rumen”

L133 – L134: It is not clear what the “sample liquid” was, and for the ratio, what 4 is and what 1 is. Revise to increase clarity.

L135: What was the substrate used? Be more specific.

L134 – L137: Was a negative control (saliva + rumen fluid + substrate) added to the in vitro run?

L137: Is there a reason why there was no replicate for run (repeating measurements on a different day)? Refer to major comments.

L144: Need to add the actual volume of culture that was centrifuged.

L152 – L155: change to “DM, crude protein, …… were analyzed according to AOAC…..)

L162 – L163: replace “the liquids after incubation” with “collected samples”

L168: Need more details and citation for the NH3-N procedure.

L212: Was data, especially bacterial relative abundance, tested for normality and homoscedacity, & transformed when appropriate, prior to analysis?

L258: delete “the”

L261: concentration

L349: change to “in vitro rumen fermentation”

L350: delete “by in vitro trial”

L350: replace “suggested” with “showed”

L352 – L353: change to “that for each 1% addition of supplemental linseed, CH4 production decreases by 4.8%”

L352: Marti et al (7) actually reported a 3.8 and not 4.8% decrease in CH4 after summarizing data from several studies

L356 – L366: The way this is discussed is problematic because there is no equivalency between the present study (batch culture) and what is reported by Martin et al (7), which was a summary of in vivo studies (continuous culture). See major comments. The discussion has to take that into account, and it would be useful to do a comparison with other in vitro studies (for an apples-to-apples comparison).

L372: change to “a feed additive could”

L372: What could possibly explain why FUM did not result in a decrease in CH4 production in the present study?

L396: replace “relate to” with “are involved in”

L397 – L398: change to “via the succinate pathway.”

L413 – L422: The decrease in digestibility in the present study vs minor impact if any of CSFA in other/cited studies [15-20], which are all in vivo, is most likely a reflection of the differences in fermentation system (batch vs continuous culture). Rumen protection is also a function of factors including rumen residence time. As rightfully stated (L453 – L456) observations will most likely be different in vivo. Use of a dual flow system (RUSITEC) would be better than batch culture, with the in vivo approach being the gold standard.

L427: replace “accounted” with “accounting for about…”

L439: replace “in” with “after”

L44: change to “considered to be one “

L448: change to “with other cellulolytic”

L449: delete “the”

L449: replace “in” with “after”

L451 – L453: This is highly speculative. Is it possible that this was an anomaly vs. an observation with biological significance? This is why replicating runs helps with providing a clearer picture.

There are indications that diet (high forage vs grain diets) could influence fermentation responses (including CH4 production) following the addition of lipids. The discussion also needs to address those potential associative effects based on substrate used.

Reviewer #2: I found nothing really critical to say about the manuscript. I agree with the statistical methods used. I agree that these results are interesting and require in-vivo experimental testing to validate the authors' findings. The use of 16s for the sequencing is probably the greatest weakness. Using shotgun metagenomic sequencing might have given better resolution than the V3-V4 amplicon method and have provided more data on the function of the microbes in the rumen microbiome.

Metagemome Assembled Genome (MAG) approaches might also have added better depth to the microbial data.

But using 16s still gave interesting results. The Shannon index suggested good diversity but the rarefaction carried out limited the sample depth to 3966 which makes me wonder whether this was too low and the supplementary data shows a bunch of stuff with LSMeans. It would be good to see the Shannon diversity graphed.

Rarefaction may have reduced the accuracy and I am going to assume the authors tested it against DESeq etc.

That aside, the methodology looked sound, the results and interpretation good. Use of V3-4 is recommended for rumen and Silva sequences also recommended. Statistically, the permutational multivariate analysis of variance (PERMANOVA) test was conducted on 999 permutations whereas our standard number we use is 10k.

Because I cant find a reason to reject this manuscript I can only point out minor potential flaws that the authors may wish to remove by explaining or providing supplementary data.

6. PLOS authors have the option to publish the peer review history of their article (what does this mean?). If published, this will include your full peer review and any attached files.

Reviewer #1: No

Reviewer #2: **Yes: **Jane Adair Mullaney

---

## [Author Response · Author response to Decision Letter 0]

8 Oct 2020

October 7, 2020

Dear Sir/Madam,

We are pleased to have an opportunity to revise our manuscript entitled, “Calcium salts of long-chain fatty acids from linseed oil decrease methane production by altering the rumen microbiome in vitro”. The reviewer’s comments are very helpful overall, and we are appreciative of such constructive feedback on our original submission. After addressing the issues raised, we feel the quality of the paper is much improved.

To facilitate your review of our revisions, the followings are point-by-point responses to the questions and comments delivered in your letter. Moreover, revised and added sentences or sections are highlighted in our revised manuscript.

【List of comments and answers】

【Reviewer 1】

The current study evaluated the impact of Ca-salts of FA from linseed oil on in vitro fermentation characteristics. Although the practical application of feeding Ca-salts as a strategy to limit CH4 emissions is questionable mostly because it would just be cheaper to feed linseed oil vs Ca-salts (typically, the cost of Ca-salts > oil > oilseeds), and that linseed oil is possibly more effective than CSFA. However, if a reduction in emissions occurs with CSFA in vivo, it would be a great thing.

Reply: As you mentioned, CSFA is 1.5 times more expensive than linseed oil. However, the reduction effect of the CSFA used in the present study was higher than that reported in a previous study (Szumacher-Strabel et al., 2004, Journal of Animal and Feed Sciences, 13: 215-218) which evaluated the effect of linseed oil using in vitro trial. Therefore, we consider that the CSFA has a potential to be a promising candidate for reduction of CH4 emission from ruminants. In order to investigate the reduction effect of the CSFA on the rumen CH4 production, we are going to conduct an in vivo study using Japanese Black cattle.

Major comments:

1. Although there was a dramatic decrease in CH4 production in the present study, it is questionable if similar observations would be made in vivo. This is related to the experimental approach. Besides being a function of the technology used (pH dependence for Ca-salts), the effectiveness of rumen-protection for products like CSFA is also influenced by factors including rumen residence time. Therefore, the in vitro system (batch culture for 48 h) is not ideal for testing the hypothesis. 

Reply: As you mentioned, the protection effect of CSFA in rumen is also influenced by rumen residence time. Therefore, the reduction effect of CSFA on CH4 evaluated in the present study may be overestimated, and it is possible not to inhibit in vivo CH4 production like in vitro. However, adding CSFA decreased CH4 after 12 h incubation, which is close to residence time of feed in rumen, indicating that CSFA is a promising candidate for reduction of CH4 emission from ruminants. We think that we need to conduct an in vivo study to evaluate the effect of CSFA in details. Therefore, we are going to perform a further in vivo study using Japanese Black cattle as mentioned above. 

One could speculate that some of the observations made in the current study (e.g., greater reduction in CH4 production with FAL and FUM compared to previous in vivo studies with linseed oil, and decrease in DM and fiber digestion with FAL and FUM, which is not typically observed with CSFA in vivo) are all a result of the closed-system (no continuous addition of substrates or passage of digesta out; fixed residence time with incubation up to 48 h).

Reply: To our knowledge, there is only one study which evaluated the effect of dietary CSFA on methane production in vivo. Kliem et al. (2019, Animal, 13.2: 309-317.) reported that CSFA from linseed and palm oil decreased CH4 in dairy cows, consistent with our results. Therefore, we think the CSFA using in the present study can probably decrease CH4 production not only in vitro but also in vivo.

With regard to a digestibility, the meanings of in vitro and in vivo digestibility are different. We can evaluate the digestibility of feed in the rumen using in vitro system (batch culture), while in vivo digestibility means the proportion of feeds digested in the total digestive tract. Therefore, we need to conduct in vivo trials to evaluate the effect of the CSFA on in vivo digestibility and we already described the necessity of further researches in Conclusion section (Line 463-465 in original manuscript). 

It is highly likely that observations made in this study might not fully translate to in vivo conditions (continuous culture). Therefore, the comparison of observations made in this study to predominantly in vivo studies in the discussion, without accounting for the differences in approach (batch vs continuous culture), is problematic. 

Reply: We agree with your comment. It is not reasonable to compare our results with in vivo studies. Therefore, we compared our results to a published paper (Szumacher-Strabel et al., 2004) which evaluated the effect of supplementary linseed oil on in vitro CH4 production, instead of Martin et al. (2010), and revised the sentence as follows:

“Thus, in this study, the percentage of CH4 reduction due to CSFA supplementation was higher than that reported by Szumacher-Strabel et al. (10.1% reduction per 1% of linseed oil addition in in vitro) [54], indicating that the CSFA used in the present study has a substantially high reduction effect on CH4 production.” (Line 363-367)

“54. Szumacher-Strabel M, Martin SA, Potkański A, Cieślak A, Kowalczyk J. Changes in fermentation processes as the effect of vegetable oil supplementation in in vitro studies. J Anim Feed Sci. 2004;13: 215–218.https://doi.org/10.22358/jafs/73843/2004” (Line 657-660)

The observations made should be interpreted with caution, and this should also be reflected in the wording of the concluding statement(s).

Reply: As you mentioned, the sentence may be an exaggerated expression because we evaluated the effect of the CSFA only in batch culture system. Therefore, we changed the sentence as follows:

“In conclusion, although further in vivo study is needed to evaluate the reduction effect on rumen CH4 production, CSFA may be a promising candidate for reduction of CH4 emission from ruminants.” (Line 38-41)

2. Given the day-to-day variation in both animal factors (e.g., nutrient intake and timing of meals, which can influence properties of the inoculum/rumen fluid) and human factors (e.g., possible errors when making reagents or purging of system to ensure anaerobic conditions, pipetting/measurement errors, etc), it is recommended that runs (days of incubation) are replicated. At least 2 incubation runs over 2 non-consecutive days are needed; in that case, the day effect is also tested in the statistical model, & is only removed if it is not significant. This potentially increases replicates (if there is no day effect), and thus experimental power, which was low in the present study, since only 3 replicates were used.

Reply: We think that only one trial for the in vitro experiment might be acceptable, since only one time in vitro fermentation was conducted to investigate the effect of feed or feed additives in many published papers such as Benchaar et al. (2007, Canadian Journal of Animal Science, 87.3: 413-419), Patra et al. (2014, Applied Microbiology and Biotechnology, 98.2: 897-905), Shen et al. (2017, Frontiers in Microbiology, 8: 1111), and Iqbal et al. (2018, Animal Nutrition, 4.1: 100-108). However, as you criticized, the low sample size and only one experiment was one of the weak points of our study. Therefore, we discussed about the point as one of the limitations of the present study in Discussion section as follows:

“One of the limitations of the present study was that the low sample size (n = 3 per treatment) with only one in vitro trial. Moreover, we evaluated the effect of CSFA using only one substrate although the effect of fat on rumen fermentation can be influenced by the concentrate and roughage ratio of feeds [91]. Therefore, further studies with an increase in sample size and substrates will be needed to increase the reliability of the effect of CSFA on CH4 production.” (Line 470-475)

The importance of replicating runs is highlighted by the observed increase in GP at 12 h of incubation when CH4 decreased. One would have greater confidence that this observation was of biological significance as discussed, rather than an error in measurement/anomaly, if the same observation is made across runs/different days of incubation.

Reply: We obtained similar results with the findings in our another trial; 20.9% increase of gas production and 40.5% decrease by adding CSFA at initial stage of ruminal incubation in batch culture system was observed (not published yet). Therefore, although we need to perform further researches in vivo or continuous culture system to increase reliability of the effect of CSFA, we consider that the results observed in the present study were not from erroneous measurements. 

3. There is no indication that a negative control (artificial saliva + rumen fluid + substrate) was included in the in vitro run. Even if it was just 1 run, a “blank correction” is important for some measures including GP as it provides more meaning data compared to just absolute values.

Reply: We prepared control (artificial saliva + rumen fluid + substrate) described as CON in the manuscript (Line 114). Because the sentence was not clear, we revised it as follows:

“The control treatment (CON) contained only substrate.” (Line 114)

We did not prepare a treatment (artificial saliva + rumen fluid) for a blank correction because we tested the effect of CSFA using only one experiment. Blank correlation has little effect on gas production in experimental treatments. In our previous studies (Sato et al., 2019; Animal Science Journal, 90(1), 90-97.; Sato et al., 2020; Livestock Science, 104217), we have prepared the control, but the value of gas production showed zero.

4. With statistical analysis it is not clear if data was tested for normality and homoscedacity prior to analysis. This is especially important with bacterial relative abundance data (proportions).

Reply: We did not test for normality and homogeneity of variance due to the lack of power of these tests for small sample size like the present study (n = 3 per treatment). Because Tukey-Kramer method may not be appropriate for analysis of bacterial relative abundance in the present study, we reanalyzed differentially abundant microbial taxa using DESeq2, which is recommended for increasing sensitivity on smaller dataset compared to traditional non-parametric tests based on Kruskal–Wallis and Wilcoxon rank-sum approaches (Weiss et al., 2015, PeerJ. 3, e1408). Based on the reanalyzed results, we revised some sentences, Tables and Figures as follows:

“The genera Ruminobacter, Succinivibrio, Succiniclasticum, Streptococcus, Selenomonas.1, and Megasphaera, which are related to propionate production, were increased (P < 0.05), while Methanobrevibacter and protozoa counts, which are associated with CH4 production, were decreased in FAH, compared with CON (P < 0.05)” (Line 32-36)

“The OTUs were rarefied to a depth of 3,966, which was the lowest sample depth, for alpha and beta diversity analysis.” (Line 207-208)

“In order to identify differentially abundant microbial taxa at the phylum and genus levels, we normalized the count matrices of taxa with a negative binomial distribution using DESeq2 [52]. Relative abundance was calculated using the normalized data, and the minor phylum and genus (average relative abundance < 1% for all treatments) were excluded from statistical analysis.” (Line 214-218)

“Data, except for Bray–Curtis dissimilarities of OTUs and abundant bacterial taxa, were analyzed using GLM procedure of Statistical Analysis System (SAS, 2008).” (Line 221-222)

“Differentially abundant bacterial taxa were identified using a negative binomial Wald test in DESeq2 [52]. The obtained p-values were corrected according to Benjamini and Hochberg procedure [53].” (Line 229-232)

We changed a subtitle in Result section to “Bacterial abundance”. (Line 313)

We revised sentences in “Bacterial abundance” section following the new results. (Line 314-330, 334-337)

We redrew Fig 4 following the new results. (Fig 4) 

We revised Fig 5 and the legend and footnotes. (Line 347-354)

We revised sentences in Discussion section according to the new results. (Line 407-419, 452-458)

We changed S1 Table and added Supporting information. (Line 800-807)

We added the reference as follows:

“52. Love MI, Huber W, Anders S. Moderated estimation of fold change and dispersion for RNA-seq data with DESeq2. Genome Biol. 2014;15: 550. https://doi.org/10.1186/s13059-014-0550-8; 53. Benjamini Y, Hochberg Y. Controlling the false discovery rate: a practical and powerful approach to multiple testing. J R Stat Soc Ser B. 1995;57: 289–300. https://doi.org/10.1111/j.2517-6161.1995.tb02031.x” (Line 651-656)

Minor comments:

L4: change to “altering the rumen”

Reply: We changed the wording following your comment. (Line 4)

L37: change to “changed the rumen”

Reply: We revised the wording accordingly. (Line 37)

L39 – L41: This is highly speculative given the approach used (batch culture) in the present study.

Reply: As stated above, we revised the sentence. (Line 38-41)

L44: change to “because it has”

Reply: We changed the wording following your comment. (Line 44)

L46: change to “Livestock are the”

Reply: We revised the wording accordingly. (Line 46)

L49: delete “as a result”

Reply: We deleted the words following your comment.

L62: change to “due to their toxic’

Reply: We changed the wording following your comment. (Line 62)

L70: change to “CSFA have generally”

Reply: We changed the wording as you suggested. (Line 70)

L91: change to “alter the microbiome”

Reply: We changed the wording following your comment. (Line 91)

L91: delete “resulting in”

Reply: We deleted the words following your comment.

L107 – L116: What was the actual experimental design? Was it a CRD? Information needs to be added

Reply: As we described in our manuscript, we collected rumen liquid from two wethers, and the liquids was mixed before using the in vitro experiment. We prepared the five treatments (CON, FAL, FAH, FUM and MON) for the experiment.

L109: replace “named” with “namely”

Reply: We replaced the word accordingly. (Line 109) 

L110: change to “of the CSFA used in this study, the…”

Reply: We changed the phrase following your comment. (Line 110)

L123 – L125: The ingredient composition should be reported on a DM basis.

Reply: We calculated the ingredient composition on a DM basis using the Standard Tables of Feed Composition in Japan (NARO 2009). Therefore, we changed the sentence as follow:

“The ingredient compositions of the concentrate were as follows: 35.2% rice bran, 54.0% rolled barley, 6.9% alfalfa meal, 3.4% soybean meal, and 0.6% vitamin-mineral premix on a DM basis calculated using the Standard Tables of Feed Composition in Japan [31].” (Line 123-126)

“31. Japan Livestock Industry Association. Standard Tables of Feed Composition in Japan. NARO, Livestock Industry Association, Tokyo. 2009.” (Line 588-589)

L129: change to “through the rumen”

Reply: We changed the word as you suggested. (Line 129)

L133 – L134: It is not clear what the “sample liquid” was, and for the ratio, what 4 is and what 1 is. Revise to increase clarity.

Reply: As you mentioned, our explanation was not clear. We changed the sentences as follows:

“The filtered sample were mixed with artificial saliva [32] in a ratio of 1:4 under anaerobic condition. The artificial saliva was sterilized by autoclaving and made anaerobic by a CO2 flushing before mixing.” (Line 132-135)

L135: What was the substrate used? Be more specific.

Reply: We replaced “substrate” with “rolled barley”. (Line 136)

L134 – L137: Was a negative control (saliva + rumen fluid + substrate) added to the in vitro run?

Reply: As stated above, we prepared the control (saliva + rumen fluid + substrate) in the present study.

L137: Is there a reason why there was no replicate for run (repeating measurements on a different day)? Refer to major comments.

Reply: As mentioned above, we could not conduct further researches because there were no cannulated wethers available for our experiment.

L144: Need to add the actual volume of culture that was centrifuged.

Reply: We centrifuged all of the remaining culture. Therefore, we replaced “Culture” with “All of the remaining culture”. (Line 145)

L152 – L155: change to “DM, crude protein, …… were analyzed according to AOAC…..)

Reply: We changed it as you suggested. (Line 154-156)

L162 – L163: replace “the liquids after incubation” with “collected samples”

Reply: We revised the phrase following your comment. (Line 164)

L168: Need more details and citation for the NH3-N procedure.

Reply: We explained the procedure more detail as follows:

“The NH3-N concentration was determined by the steam distillation in a micro-Kjeldahl system (Kjeltec 2300, Foss Japan Ltd., Tokyo, Japan). Briefly, 3 mL of the supernatant after incubation was distilled with NaOH and the NH3-N was trapped in a boric acid solution. Then, the solution was titrated with 0.1 N H2SO4 to determine NH3-N concentration.” (Line 169-173)

L212: Was data, especially bacterial relative abundance, tested for normality and homoscedacity, & transformed when appropriate, prior to analysis?

Reply: As stated above, we did not test for normality and homogeneity of variance. We normalized our data with a negative binomial distribution using DESeq2 and reanalyzed.

L258: delete “the”

Reply: We deleted the word following your comment.

L261: concentration

Reply: We revised it accordingly. (Line 269)

L349: change to “in vitro rumen fermentation”

Reply: We changed the word as you suggested. (Line 356)

L350: delete “by in vitro trial”

Reply: We deleted the word following your comment.

L350: replace “suggested” with “showed”

Reply: We replaced the word accordingly. (Line 357)

L352 – L353: change to “that for each 1% addition of supplemental linseed, CH4 production decreases by 4.8%”

Reply: We changed the phrase following your comment. (Line 359-360)

L352: Martin et al. (7) actually reported a 3.8 and not 4.8% decrease in CH4 after summarizing data from several studies.

Reply: As you mentioned, Martin et al. (2010) reported that 3.8% CH4 was decreased with 1% addition of supplemental fat decreased. However, they also described that CH4 was decreased by 4.8% with supplementary 1% linseed. 

L356 – L366: The way this is discussed is problematic because there is no equivalency between the present study (batch culture) and what is reported by Martin et al (7), which was a summary of in vivo studies (continuous culture). See major comments. The discussion has to take that into account, and it would be useful to do a comparison with other in vitro studies (for an apples-to-apples comparison).

Reply: As stated above, instead of Martin et al. (2010), we compared our results to a published paper (Szumacher-Strabel et al., 2004), which evaluated the effect of supplementary linseed oil on in vitro CH4 production and revised the sentences.

L372: change to “a feed additive could”

Reply: We changed the wording following your comment. (Line 380)

L372: What could possibly explain why FUM did not result in a decrease in CH4 production in the present study?

Reply: We consider that CH4 reduction effect of FUM was low (only 6.7% reduction) because rolled barley was used as the substrate in the present study. García-Martínez et al. (2005) reported that adding fumarate to batch culture under a low-forage substrate condition have less CH4 reduction effect compared with a high-forage substrate condition. Considering their study, we added the explanation for the reason why fumarate had little CH4 reduction effect as follows:

“In the present study, supplementary fumarate did not reduce CH4 production probably because rolled barley was used as the substrate. García-Martínez et al. [56] reported that adding fumarate to batch culture under a low-forage substrate condition have less CH4 reduction effect compared with a high-forage substrate condition.” (L 383-387)

“56. García-Martínez R, Ranilla MJ, Tejido ML, Carro MD. Effects of disodium fumarate on in vitro rumen microbial growth, methane production and fermentation of diets differing in their forage: concentrate ratio. Br J Nutr. 2005;94: 71–77. https://doi.org/10.1079/BJN20051455” (Line 664-667)

L396: replace “relate to” with “are involved in”

Reply: We replaced the words as you suggested (Line 408)

L397 – L398: change to “via the succinate pathway.”

Reply: We changed the words accordingly (Line 409-410)

L413 – L422: The decrease in digestibility in the present study vs minor impact if any of CSFA in other/cited studies [15-20], which are all in vivo, is most likely a reflection of the differences in fermentation system (batch vs continuous culture). Rumen protection is also a function of factors including rumen residence time. As rightfully stated (L453 – L456) observations will most likely be different in vivo. Use of a dual flow system (RUSITEC) would be better than batch culture, with the in vivo approach being the gold standard.

Reply: We agree with your comment. As you mentioned, the results may differ between in vivo and vitro. In our previous study, we observed the difference of the results between in vivo and vitro (Sato et al., 2020, Livestock Science, 104217). Therefore, we consider that we need to conduct further researches using in vivo trial. Although we already described the necessity of further researches in Conclusion section, we changed the sentences as follows:

“These findings present CSFA as a promising candidate for reduction of CH4 emission from ruminants. However, some differences of the observation were reported between in vivo and vitro [92]. Therefore, future studies are needed to confirm the in vivo effect of dietary CSFA on CH4 production, productivity, rumen microbiome, and digestibility, and to determine the optimal amount of CSFA in a diet for ruminants.” (Line 479-483) 

“92. Sato Y, Nakanishi T, Wang L, Oishi K, Hirooka H, Kumagai H. In vitro and in vivo evaluations of wine lees as feeds for ruminants: Effects on ruminal fermentation characteristics, nutrient digestibility, blood metabolites and antioxidant status. Livest Sci. 2020; 104217. https://doi.org/10.1016/j.livsci.2020.104217” (Line 791-795)

L427: replace “accounted” with “accounting for about…”

Reply: We replaced it following your comment. (Line 437-438)

L439: replace “in” with “after”

Reply: We changed the word accordingly. (Line 449)

L44: change to “considered to be one “

Reply: We changed it as you suggested. (Line 455)

L448: change to “with other cellulolytic”

Reply: We changed the wording following your comment. (Line 459)

L449: delete “the”

Reply: We deleted it.

L449: replace “in” with “after”

Reply: We revised the word as you suggested. (Line 460)

L451 – L453: This is highly speculative. Is it possible that this was an anomaly vs. an observation with biological significance? This is why replicating runs helps with providing a clearer picture.

Reply: As stated above, we obtained similar results with our findings in an another trial (data not published). Therefore, although we need to perform further researches in vivo or continuous culture system to increase reliability of the effect of CSFA, we consider that the results observed in the present study were not anomaly. 

There are indications that diet (high forage vs grain diets) could influence fermentation responses (including CH4 production) following the addition of lipids. The discussion also needs to address those potential associative effects based on substrate used.

Reply: As you mentioned, the effect of fat on rumen fermentation is influenced by a kind of substrates. So, we added the sentence in Discussion section as follows:

“One of the limitations of the present study was that the low sample size (n = 3 per treatment) with only one in vitro trial. Moreover, we evaluated the effect of CSFA using only one substrate although the effect of fat on rumen fermentation can be influenced by the concentrate and roughage ratio of feeds [91]. Therefore, further studies with an increase in sample size and substrates will be needed to increase the reliability of the effect of CSFA on CH4 production” (Line 470-475)

“91. Bayat AR, Ventto L, Kairenius P, Stefanski T, Leskinen H, Tapio I, et al. Dietary forage to concentrate ratio and sunflower oil supplement alter rumen fermentation, ruminal methane emissions, and nutrient utilization in lactating cows. Transl Anim Sci. 2017;1: 277–286. https://doi.org/10.2527/tas2017.0032” (Line 787-790)

【Reviewer 2】

I found nothing really critical to say about the manuscript. I agree with the statistical methods used. I agree that these results are interesting and require in-vivo experimental testing to validate the authors' findings. The use of 16s for the sequencing is probably the greatest weakness. Using shotgun metagenomic sequencing might have given better resolution than the V3-V4 amplicon method and have provided more data on the function of the microbes in the rumen microbiome.

Metagemome Assembled Genome (MAG) approaches might also have added better depth to the microbial data. But using 16s still gave interesting results.

Reply: Thank you for your helpful suggestion. We fully understand that shotgun sequence is better to clarify the function of the microbes in the rumen. We plan to conduct in vivo experiment to investigate the effect of dietary CSFA on methane production and the rumen microbiome in Japanese Black cattle. We are going to use shotgun sequence in the experiment according to your suggestion.

The Shannon index suggested good diversity but the rarefaction carried out limited the sample depth to 3966 which makes me wonder whether this was too low and the supplementary data shows a bunch of stuff with LSMeans. 

Reply: As you mentioned, the sample depth in our study was small. However, there have been many reports using less sampling depth than 3966. For example, Lourenco et al. (2019; Frontiers in Microbiology, 10: 1131), Acosta et al. (2019; PloS ONE, 15.2: e0228560) and Rubanov et al. (2019; Scientific Reports, 9.1: 1-8) used 3484, 3664, and 2500 as the sampling depth, respectively.

It would be good to see the Shannon diversity graphed.

Reply: Visualization of alpha diversity is an effective method to express the result clearly. So, we added the figure (Fig 1) and the legend and footnotes (Line 293-299), and deleted Table 4. Additionally, we renumbered all figures (Fig 1-5). 

Rarefaction may have reduced the accuracy and I am going to assume the authors tested it against DESeq etc.

Reply:　As you mentioned, rarefaction might reduce the accuracy. Therefore, according to your suggestion, we normalized the count matrices of taxa with a negative binomial distribution using DESeq2 and tested Shannon diversity. The value (mean ± SE) in CON, FAL, FAH, FUM and MON were 6.23±0.04, 5.71±0.07, 4.95±0.09, 5.61±0.07 and 5.12±0.05, respectively. The results were almost equivalent to those normalized using rarefaction (CON, 6.25±0.07; FAL, 5.73±0.08; FAH, 4.96±0.11; FUM, 5.62±0.07; MON, 5.22±0.07). The result indicates that the rarefaction had no effect on the accuracy in the present study.

That aside, the methodology looked sound, the results and interpretation good. Use of V3-4 is recommended for rumen and Silva sequences also recommended. Statistically, the permutational multivariate analysis of variance (PERMANOVA) test was conducted on 999 permutations whereas our standard number we use is 10k.

Reply: We agree with your comment. Using V3-V4 variable region and Silva sequences is reasonable to determine the rumen microbiota. As you mentioned, 9999 permutation is better to estimate p-value accurately. So, we conducted PERMANOVA using 9999 permutations, and the p-value was less than 0.05 like the result of PERMANOVA using 999 permutations. 

We changed the words “999 permutations” to “9999 permutations” in Materials and Methods section. (Line 229)

Because I can’t find a reason to reject this manuscript I can only point out minor potential flaws that the authors may wish to remove by explaining or providing supplementary data.

Reply: Thank you for giving us the insightful feedbacks to strengthen our manuscript.

---

## [Decision Letter · Decision Letter 1]

28 Oct 2020

Calcium salts of long-chain fatty acids from linseed oil decrease methane production by altering the rumen microbiome in vitro

PONE-D-20-14439R1

Dear Dr. Kumagai,

We’re pleased to inform you that your manuscript has been judged scientifically suitable for publication and will be formally accepted for publication once it meets all outstanding technical requirements.

Kind regards,

Brenda A Wilson, Ph.D.

Academic Editor

PLOS ONE

Additional Editor Comments (optional):

Reviewers' comments:

Reviewer's Responses to Questions

**Comments to the Author**

1. If the authors have adequately addressed your comments raised in a previous round of review and you feel that this manuscript is now acceptable for publication, you may indicate that here to bypass the “Comments to the Author” section, enter your conflict of interest statement in the “Confidential to Editor” section, and submit your "Accept" recommendation.

Reviewer #1: All comments have been addressed

Reviewer #2: All comments have been addressed

2. Is the manuscript technically sound, and do the data support the conclusions?

Reviewer #1: (No Response)

Reviewer #2: Yes

3. Has the statistical analysis been performed appropriately and rigorously? 

Reviewer #1: (No Response)

Reviewer #2: Yes

4. Have the authors made all data underlying the findings in their manuscript fully available?

Reviewer #1: (No Response)

Reviewer #2: Yes

5. Is the manuscript presented in an intelligible fashion and written in standard English?

Reviewer #1: (No Response)

Reviewer #2: Yes

6. Review Comments to the Author

Reviewer #1: (No Response)

Reviewer #2: (No Response)

7. PLOS authors have the option to publish the peer review history of their article (what does this mean?). If published, this will include your full peer review and any attached files.

Reviewer #1: No

Reviewer #2: **Yes: **Jane Adair Mullaney

---

## [Editor Report · Acceptance letter]

30 Oct 2020

PONE-D-20-14439R1 

Calcium salts of long-chain fatty acids from linseed oil decrease methane production by altering the rumen microbiome *in vitro*

Dear Dr. Kumagai:

I'm pleased to inform you that your manuscript has been deemed suitable for publication in PLOS ONE. Congratulations! Your manuscript is now with our production department. 

Kind regards, 

on behalf of

Dr. Brenda A Wilson 

Academic Editor

PLOS ONE